# Novel Pathogenic Variant of the *TRRAP* Gene Detected in a Hungarian Family with Autosomal Dominant Non-Syndromic Hearing Loss

**DOI:** 10.3390/ijms26041583

**Published:** 2025-02-13

**Authors:** Nikoletta Nagy, Ágnes Szalenko-Tőkés, Margit Pál, Barbara Anna Bokor, Roland Nagy, János András Jarabin, László Róvó, Márta Széll

**Affiliations:** 1Department of Medical Genetics, University of Szeged, 6720 Szeged, Hungary; pal.margit@med.u-szeged.hu (M.P.); bokor.barbara.anna@med.u-szeged.hu (B.A.B.); szell.marta@med.u-szeged.hu (M.S.); 2HUN-REN-SZTE Functional Clinical Genetics Research Group, University of Szeged, 6726 Szeged, Hungary; 3Department of Oto-Rhino-Laryngology and Head-Neck Surgery, University of Szeged, 6720 Szeged, Hungary; szalenko-tokes.agnes@med.u-szeged.hu (Á.S.-T.); nagy.roland@med.u-szeged.hu (R.N.); jarabin.janos.andras@med.u-szeged.hu (J.A.J.); office.orl@med.u-szeged.hu (L.R.)

**Keywords:** hearing loss, whole exome sequencing, novel, variant, missense, TRRAP

## Abstract

Autosomal dominant non-syndromic hearing loss (ADNSHL) is a genetically heterogenic condition. The transformation/transcription domain associated protein (*TRRAP*) gene has been recently associated with ADNSHL, and only four variants of the gene have so far been reported in this disease. Here, we report on a Hungarian ADNSHL family in which the affected individuals exhibited sensorineural hearing loss with similar clinical symptoms, including initial impaired high frequencies that subsequently affected speech and lower frequencies. Whole exome sequencing and screening of the shared genetic variants of the affected individuals was performed. Our results revealed a novel heterozygous missense variant (NM_001244580.2, c.5360A>G, p.Lys1787Arg) in the *TRRAP* gene. This variant is completely co-segregated with hearing impairment. It is present in a heterozygous form in the affected mother and daughter but not carried by any unaffected family members. This study highlights the importance of elucidating the germline genetic background of ADNSHL, which may help to predict individual risk and the risk of family members. This will improve prevention, screening, and therapeutic measures for each patient and hearing loss-prone families.

## 1. Introduction

Genetic hearing loss is a common congenital disease with a prevalence of 1/1000 in humans [1,2]. Approximately 70–80% of genetic hearing loss is non-syndromic and shows autosomal dominant, autosomal recessive, X-linked, or mitochondrial inheritance [1,2]. The genetic variants of more than 80 genes are known to be associated with non-syndromic hearing loss. The most frequently mutated genes include *GJB2*, *SLC26A4*, *MYO15A*, *OTOF*, *TMC1*, *CDH23,* and *GJB6*. With respect to autosomal dominant non-syndromic hearing loss (ADNSHL), hearing loss may occur at an advanced age and presents in variable ways and severities. Its genetic background is very heterogenic, with 44 disease-causing genes that have been found to be associated with its development thus far (http://hereditaryhearingloss.org/, accessed on 13 July 2024) [3,4]. Despite the high number of previously identified genes, a large portion of ADNSHL has not been genetically defined, which is a significant issue for this disease [4].

Among these 44 genes, there is the recently reported disease-causing gene, the transformation/transcription domain associated protein (*TRRAP*) [5]. The encoded TRRAP protein is an evolutionary conserved member of the phosphoinositide 3-kinase-related kinases (PIKK) family [6,7]. TRRAP is a component of many histone acetyltransferase (HAT) complexes and plays a role in transcription and DNA repair by recruiting HAT complexes to chromatin [8,9]. TRRAP also has a significant role in hearing development. In trrap knockdown and knockout, zebrafish had reduced lateral line neuromasts, decreased number of hair cells per neuromast, and abnormal stereocilia on the hair cells compared with WT zebrafish [5].

In humans, two rare diseases with autosomal dominant inheritance have been associated with *TRRAP*: the ADNSHL (OMIM 618778) and the developmental delay with or without dysmorphic facies and autism (OMIM 618454). With respect to ADNSHL, only four disease-causing *TRRAP* variants—three missense (NM_001244580.2, p.Arg171Cys, p.Asp394Asn, and p.Glu2750Asp) and one frameshift (p.Pro509fs)—have been linked with the disease [5]. Nearly all previously reported variants affect the same functional domain of the TRRAP protein, including the p.Arg171Cys, p.Asp394Asn, and p.Pro509fs variants, which all affect the Tra1 HEAT repeat central region domain (position from 18 to 531 amino acids), whereas the p.Glu2750Asp variant does not affect any known functional domains of the protein [5].

Here, we report on a Hungarian ADNSHL family in which the affected individuals show hearing loss with very similar unique clinical characteristics. Whole exome sequencing (WES) was performed, and the shared genetic variants of the affected individuals were assessed to establish the genetic background of their hearing loss.

## 2. Results

### 2.1. Audiological Investigations

Audiological tests were performed, and the affected individuals showed sensorineural hearing loss with similar clinical features (Figure 1).

In the case of the mother (II/2), the pure-tone audiometry (PTA) results showed significantly impaired high-tone perception in the 2–8 kHz range, whereas the low-tone (0.125–0.5 kHz) range perception was spared (Figure 2).

The hearing threshold at 1 kHz was mildly impaired. Tympanometry revealed normal middle ear ventilation with −3 daPa and 38 daPa pressures (“A” type tympanogram). The acoustic reflex threshold (ART) was absent at 500 and 1000 Hz up to 100 dB stimulus intensity. Regarding the distortion-product otoacoustic emission (DPOAE), at 70/60 dB SPL stimulus intensity, the DP-grams showed absent otoacoustic emissions in both ears (Figure 3).

Click-evoked–auditory brain stem response audiometry (click-ABR) and auditory steady-state response audiometry (ASSR) measurements were performed (Figure 4).

At 90 dB nHL stimulus intensity from the right ear, the Vth wave was indefinably present. In contrast, from the left ear, the Vth wave was absent. The objective hearing thresholds measured at frequencies of 0.5, 1, 2, and 4 kHz were consistent with the PTA subjective results (right ear: 15, 20, 100, and 100 dB corHL; left ear: 15, 35, 75, and 80 dB corHL).

For the daughter (III/2), the PTA results showed a significantly impaired high-tone perception in the 2–8 kHz range (Figure 2), whereas the low-tone (0.125–1 kHz) range perception was normal. Tympanometry proved normal middle ear ventilation with 0 daPa and 0 daPa pressure (“A” type tympanogram). The ART was 500 and 2000 Hz, while absent at 4 kHz. At 70/60 dB SPL stimulus intensity, the DP-grams showed nearly absent otoacoustic emissions in both ears (Figure 3). At 90 dB nHL stimulus intensity, only the Vth waves were defined and reproducible on both sides. The objective hearing thresholds, measured at frequencies of 0.5, 1, 2, and 4 kHz, were consistent with the PTA subjective results (right ear: 5, 30, 65, and 65 dB corHL; left ear: 30, 55, 75, and 80 dB corHL) (Figure 4). MRI did not reveal any abnormalities in the inner ears of the affected mother and daughter (Figure 5).

The audiograms for both individuals showed significantly impaired high-tone perception, whereas the low-tone perception was spared with a “ski-slope-type curve”. Based on the medical history of the subjects, their hearing loss may have developed at the same time. As the hearing threshold of the II/2 mother deteriorated only slightly over time, this slow progression may continue for decades. This is important for long-term hearing rehabilitation, as conventional hearing assistive solutions appear to provide stable and good rehabilitative effects over the long term. Based on the available data, an indication for cochlear implantation was not supported.

### 2.2. Genetic Investigations

A genetic examination of the affected family was initiated. WES identified a novel (c.5360A>G, p.Lys1787Arg) rare germline heterozygous missense variant in the 38th exon of the *TRRAP* gene NM_001244580.2 (Figure 6) located at 7q22.1.

This variant was present in the two affected patients (II/2 mother and III/2 daughter) and not found in any unaffected family members (I/2 grandmother, II/1 father, and III/1 daughter). Regarding the variant frequency data in different populations, this variant shows extremely low frequency in gnomAD population databases (PM2). The pathogenicity of this variant is further supported by its complete cosegregation with the phenotype (PP1). Based on the ACMG variant classification guideline, this variant is classified as a VUS.

The region of the variant on the TRRAP protein exhibits high evolutionary conservation (Figure 7) (Aminode, http://www.aminode.org/search, accessed on 21 August 2024).

The PhyloP100way score, based on multiple alignments of 99 vertebrate genome sequences to the human genome, of this variant is 9.234 (https://varsome.com/variant/hg38/rs1333016292?annotation-mode=germline, accessed on 21 August 2024).

In silico functional predictions using SHIFT, Mutation Taster, and CADD suggested that the newly identified missense variant may have a putative disease-causing role in the development of the observed hearing loss, while Polyphen2, EVE, REVEL, and AlphaMissense suggest a benign effect (Figure 8).

Regarding the encoded TRRAP protein (UniProt Q9Y4A5), the detected missense variant affects the Tra1 HEAT repeat ring region (located from amino acid 541 to amino acid 2011) (Figure 8) (Protein Data Bank, https://www.rcsb.org/3d-sequence/7KTR?asymId=A, accessed on 28 January 2025). The Tra1 HEAT repeat domains are not only involved in mediating interactions with other proteins but also contribute to the dynamic nature of these interactions. The flexibility inherent in the HEAT repeat domains allows conformational changes that are necessary for binding different partners under varying cellular conditions (https://www.ebi.ac.uk/interpro/, accessed on 28 January 2025).

Using 3D modeling software (SWISS-MODEL, https://swissmodel.expasy.org/interactive, accessed on 28 January 2025), the homology between the 3D structure of a 31-amino-acid-long region of the Tra1 HEAT repeat ring domain containing wild-type sequence or the detected missense variant were compared to each other. The 3D protein modeling demonstrated that the p.Lys1787 and the p.Arg1787 are different in their side chain structures.

## 3. Discussion

Using WES, we identified a novel heterozygous missense variant (p.Lys1787Arg) of the TRRAP gene that shows complete cosegregation with the ADNSHL phenotype in a Hungarian family. Interestingly, *TRRAP* has been recently linked to this phenotype (2019), and so far, only four variants of the gene have been associated with this condition [5].

The identified novel variant (p.Lys1787Arg) affects the Tra1 HEAT repeat ring region domain of the TRRAP protein, which consists of alpha solenoid repeats that form a ring region [10]. Comparison of the 3D structures of p.Lys1787 and p.Arg1787 suggested differences in their side chain structures, which can lead to different binding affinities and interaction dynamics with other proteins. Tra1 domains are important for the recruitment and activation of SAGA and NuA4 complexes [11,12]. Previously, it was suggested that the identified genetic variants affect this function of the protein [5]. We assume that similarly to this, the novel variant may also affect this function. Further studies are needed to examine this putative mechanism and other unidentified mechanism(s), which may also explain how the ADNSHL-causing variants of the *TRRAP* gene contribute to disease development.

Due to the severe grade of high-frequency sensorineural hearing impairment in the 2–4 kHz range, a differential diagnosis between the cochlear vs. retrocochlear origin requires more complex diagnostic procedures, such as MRI. Although the MRI examination did not reveal any abnormalities of the inner ear, the audiological findings partially support (on the contrary, do not exclude) the cochlear origin of hearing loss, and the cochlea may be the site of the lesion. This correlates well with the results of the trrap knockdown and knockout zebrafish study: the observed reduced lateral line neuromasts, the decreased number of hair cells per neuromast, and the abnormal stereocilia on the hair cells of the animals suggest relevant functional losses that resemble the diseased phenotype [5].

Our study further demonstrates that the genetic background of ADNSHL is highly complex, widens the spectrum of the known disease-associated variants in this condition, and confirms that *TRRAP* is truly a candidate gene in ADNSHL. Our results further expand our knowledge and broaden the genetic landscape of ADNSHL in the Hungarian population [13]. From the perspective of the clinical geneticist, we conclude that WES is an effective approach for establishing the genetic background of ADNSHL. Our study also highlights that the identification of the precise genetic background is of clinical significance because it estimates the risk of the individual family members to ADNSHL. In the long run, these types of studies may contribute to the development of preventive screening and, probably, in the future, effective therapies for patients and hearing loss-prone families. Nonetheless, the underlying mechanism through which this newly detected variant contributes to the development of this unique phenotype requires further study.

## 4. Materials and Methods

### 4.1. Patients and Samples

A Hungarian pedigree with ADNSHL was enrolled (Figure 1 pedigree). The onset of the hearing loss was in the childhood of both patients: around 8 years of age in the affected mother and 7 in the daughter. A genetic analysis was performed on two affected family members (II/2 mother and III/2 daughter) and three clinically unaffected family members (I/2 grandmother, II/1 father, and III/1 daughter). A detailed clinical workup was performed. To test the auditory system PTA, impedance audiometry (tympanometry and ART), DPOAE, click-ABR, and ASSR measurements were performed. The affected individuals showed sensorineural hearing loss with similar clinical features. High frequencies were first impaired, followed by speech and lower frequencies. From the audiological and otoneurological examination, the audiogram (Figure 1) showed sensorineural hearing loss that primarily affected high frequencies. On examination, vestibular dysfunction and neurological abnormalities were not observed in the patients. All enrolled subjects had normal ear anatomy and none had any prior ear surgery. A magnetic resonance imaging (MRI) examination was also performed (Figure 1).

After genetic counseling and written informed consent were obtained, peripheral blood samples were collected, and genomic DNA was isolated using QIAGEN kits (Hilden, Germany). Genetic testing for ADNSHL was conducted according to recommendations published online (http://hereditaryhearingloss.org/, accessed on 10 January 2025). This study was approved by the Hungarian National Public Health Centre and was conducted according to the Helsinki guidelines.

### 4.2. Whole Exome Sequencing

Patients’ genotypes were determined using targeted next-generation sequencing (NGS). Libraries were prepared using the SureSelectQXT Reagent Kit (Agilent Technologies, Santa Clara, CA, USA). Pooled libraries were sequenced on an Illumina NextSeq 550 NGS platform using a 300-cycle Mid Output Kit v2.5 (Illumina, Inc., San Diego, CA, USA). Adapter-trimmed and Q30-filtered paired-end reads were aligned to the hg19 human reference genome using the Burrows–Wheeler Aligner (BWA). Duplicates were marked using the Picard software package. The Genome Analysis Toolkit (GATK) was used for variant calling (BaseSpace BWA Enrichment Workflow v2.1.1. with BWA 0.7.7-isis-1.0.0, Picard: 1.79 and GATK v1.6-23-gf0210b3).

Sequencing revealed that the mean on-target coverage was 71× per base with an average percentage of targets covered that were greater or equal to 30×, respectively. Variants passed through the GATK filter were used for downstream analysis and annotated with the ANNOVAR software tool (https://www.openbioinformatics.org/annovar/annovar_download_form.php, accessed on 10 January 2023). Single-nucleotide polymorphism testing was performed as follows: high-quality sequences were aligned with the human reference genome (GRCh37/hg19) to detect sequence variants, which were analyzed and annotated. The variants were filtered according to read depth, allele frequency, and prevalence reported in genomic variant databases, such as ExAc (v.0.3) and Kaviar. Variant prioritization tools (PolyPhen-2, SIFT, LRT, Mutation Assessor) were used to predict the functional impact of the mutation. These predictions were checked in Ensemble Genome Browser (https://www.ensembl.org/, accessed on 10 July 2024. We interpreted the sequencing results using the Franklin Genoox website. VarSome and Franklin bioinformatic platforms (https://franklin.genoox.com, accessed on 10 July 2023) were used based on the guidelines of the American College of Medical Genetics and Genomics. The candidate variants were confirmed by bidirectional capillary Sanger sequencing. Candidate variants are listed in the Appendix A.

## 5. Conclusions

A novel pathogenic germline heterozygous missense variant (c.5360A>G, p.Lys1787Arg) was identified in the *TRRAP* gene (NM_001244580.2) of a Hungarian family that presented with ADNSHL. This gene has been recently linked to ADNSHL, and its disease-causing variants (*n* = 4) have only been found in Chinese patients. This is the second study confirming an association between *TRRAP* and ADNSHL. Our study also highlights the importance of establishing the genetic background of ADNSHL and demonstrates that considering its complexity. WES is a straightforward and effective approach to genetic disease screening. Our study provides further insight into ADNSHL and improves the estimation of the risk to individuals and their family members.

## Figures and Tables

**Figure 1 ijms-26-01583-f001:**
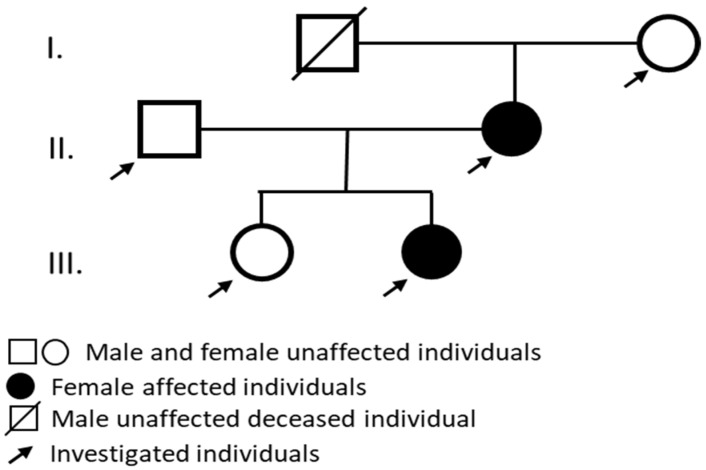
On the pedigree, Hungarian patients II/2. and III/2. are suffering from ADNSHL.

**Figure 2 ijms-26-01583-f002:**
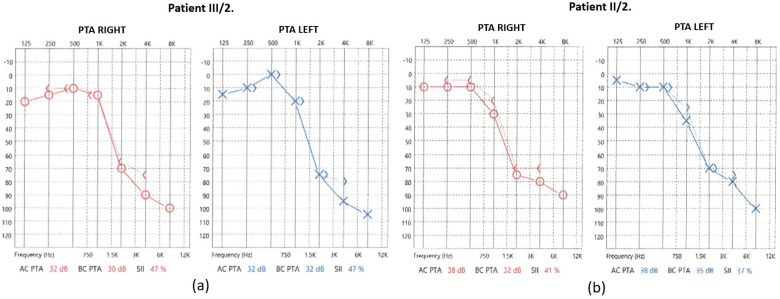
PTA results of (**a**) patient III/2 and (**b**) patient II/2.

**Figure 3 ijms-26-01583-f003:**
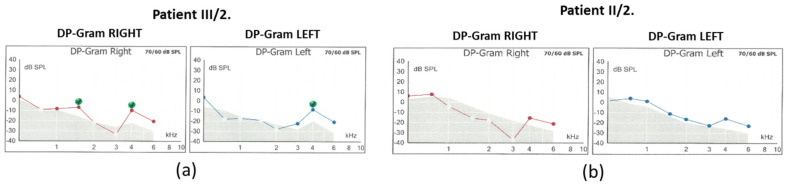
DP-grams of (**a**) patient III/2 and (**b**) patient II/2.

**Figure 4 ijms-26-01583-f004:**
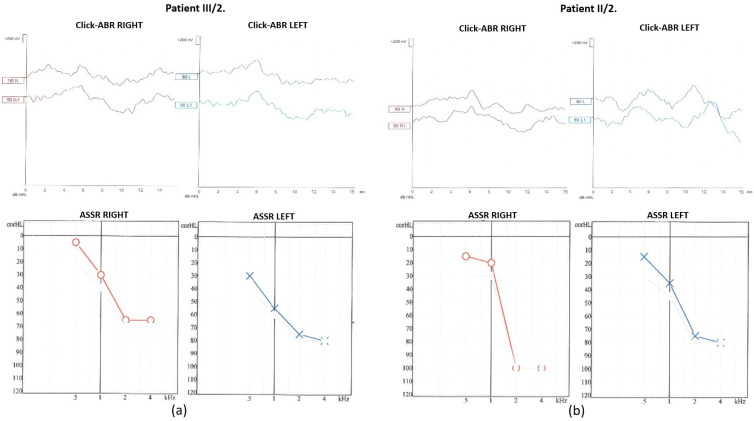
Click-ABR and ASSR results of (**a**) patient III/2 and (**b**) patient II/2.

**Figure 5 ijms-26-01583-f005:**
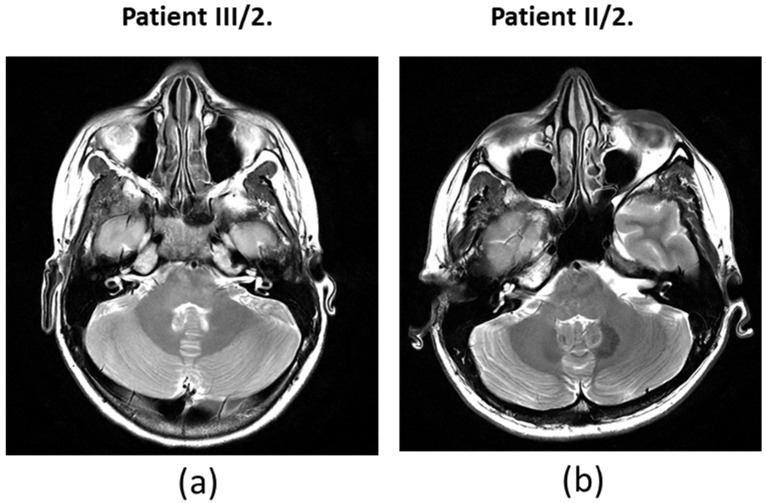
MRI of (**a**) patient III/2 and (**b**) patient II/2.

**Figure 6 ijms-26-01583-f006:**
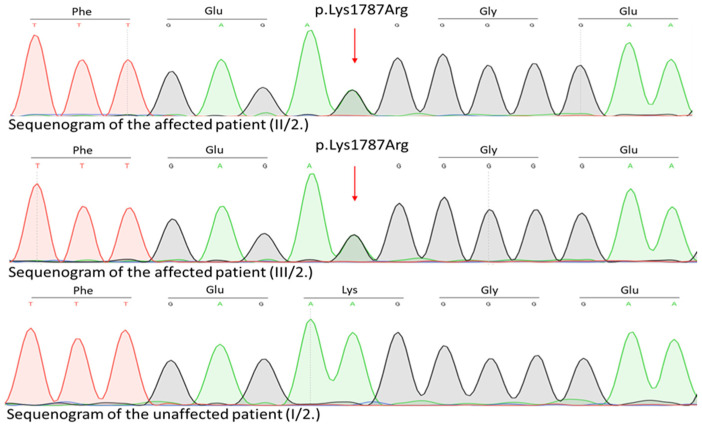
Location of the newly identified *TRRAP* variant; sequenograms for the affected patients and unaffected family members.

**Figure 7 ijms-26-01583-f007:**
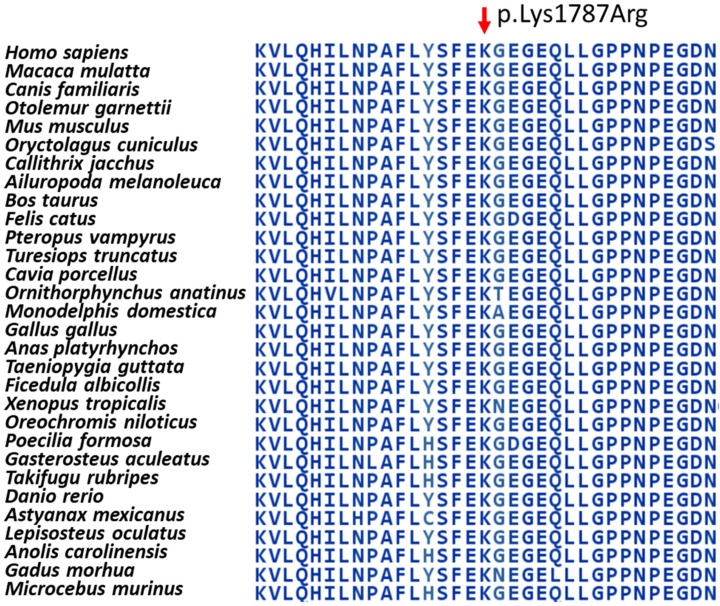
Evolutionary conservation of the newly identified *TRRAP* variant.

**Figure 8 ijms-26-01583-f008:**
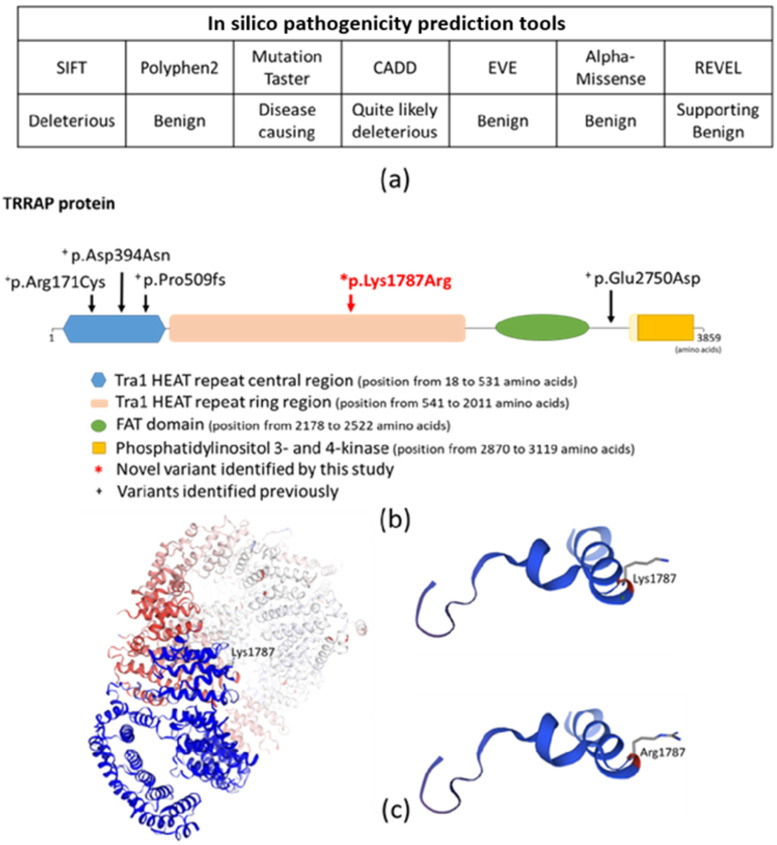
(**a**) Pathogenicity prediction of the novel variant using in silico tools and (**b**) location of the newly identified variant on the TRRAP protein. (**c**) In silico prediction of the effect of the missense variant on the 3D structure of the TRRAP protein (https://swissmodel.expasy.org, accessed on 28 January 2025).

## Data Availability

The data presented in this study are available on request from the corresponding author. The data are not publicly available because they are genetic data.

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
