# Peer review of "Novel Pathogenic Variant of the *TRRAP* Gene Detected in a Hungarian Family with Autosomal Dominant Non-Syndromic Hearing Loss"

_ijms, 2025, doi:10.3390/ijms26041583_

Round 1

Reviewer 1 Report

Comments and Suggestions for Authors

Nagy et al identified a novel heterozygous missense variant (NM_001244580.2, c.5360A>G, p.(Lys1787Arg) in the TRRAP gene. Here are a few suggestions. 

· While analyzing the Exome analysis, which cut-off value or parameters were used for minor allelic frequency based on the autosomal dominant model? After filtration of the exome data, how many heterozygous variants were left? Did you identify any other variants in known hearing loss genes?

· Please provide the data of filtered variants in the supplementary table. Here is one the papers as an example where the exome filtered variants data was added as a supplementary table (PMID: 29971487).

· Please add the REVEL score of this missense variant c.5360A>G p.(Lys1787Arg).

· Did you observe vestibular dysfunction in the patient? What is the age onset of hearing loss in affected individuals, did you observe any other neurological abnormality in the patients? please add in the manuscript.

· There is no previously reported variant in the Tra1 HEAT repeat ring region domain of the protein. Please describe in the discussion how important this specific domain is for the protein functioning.

· Although the Insilco tools suggest the variant is disease-causing, there are no in vitro experiment details provided in this study that further describe the pathogenicity of this variant.

· The minor allelic frequency (MAF) of this variant in gnomAD (v2.1) is 0.00001463 (4 hets) and in gnomAD (v4.1) is 0.000009352 (15 hets), and significantly high in the European population, this MAF for autosomal dominant variants usually considered not much significant for autosomal dominant inheritance. So, is this MAF still questionable for this variant?

· Please perform the 3D modeling of this missense variant to show how it is affecting the protein structure.

· Line 136, Please clarify the word “leaning pathogenic VUS”? and if it is VUS please write the complete word such as a variant of uncertain significance (VUS). There is no such term that exists in ACMG guidelines. This can either be “VUS” or “Likely pathogenic”. To classify the missense variant as likely pathogenic, at least two moderate and two supporting pieces of evidence must be provided. In this manuscript, the claim of likely pathogenic is “FALSE” for this missense variant, c.5360A>G, p.(Lys1787Arg) as only two criteria are provided in this manuscript including PM2 and PP1. So PM2 and PP1 both don’t fulfil the requirement of ACMG guidelines for missense variants to be classified as likely pathogenic. However, this variant is still a VUS. 

Reviewer 2 Report

Comments and Suggestions for Authors

Dear Authors,

Thank you for your article. The TRRAP gene is indeed a fascinating topic, with two conditions associated with pathogenic variants in this gene: deafness and developmental delay, with or without dysmorphism and autism.

Introduction
The introduction is well-structured and provides sufficient background information.

Results
This section represents the main issue with your article. You report a variant that is classified as a variant of uncertain significance (VUS). While I understand that applying ACMG guidelines can be challenging in rare diseases, particularly when few variants have been reported, this limitation must be addressed. For publication in a high-impact journal, conducting functional studies and in vivo analyses would be necessary to strengthen your findings. The evidence you present, particularly the presence of the affected mother, does suggest that the variant is likely pathogenic. However, additional experimental data would be required to conclusively support this classification.

Conclusion
The remainder of the article is well-written and presents its arguments clearly. Nevertheless, the primary concern is that the variant in question is not officially classified as pathogenic, which weakens the overall impact of the findings.

Round 2

Reviewer 1 Report

Comments and Suggestions for Authors

Thanks for the response. The author needs to add the details of the 3D modeling in the manuscript in the results and also discuss those results in the discussion section.  

Author Response

Reply to Reviewer 1

Thanks for the response. The author needs to add the details of the 3D modeling in the manuscript in the results and also discuss those results in the discussion section.  

Reply: Thank you for your suggestion. The details of the 3D modeling has been added to the results (lines: 164-168) and to the discussion (lines: 176-178) in the revised manuscript.

Reviewer 2 Report

Comments and Suggestions for Authors

Dear authors, 

High-impact journals often prefer comprehensive studies that include both variant identification and functional validation. If the variant is published alone, its biological significance may be questioned. However, since in vivo studies typically take longer, publishing the variant first can help establish priority while allowing time for functional validation. I understand and respect your decision to publish this novel variant first.

Author Response

Reply to Reviewer 2

Dear authors, 

High-impact journals often prefer comprehensive studies that include both variant identification and functional validation. If the variant is published alone, its biological significance may be questioned. However, since in vivo studies typically take longer, publishing the variant first can help establish priority while allowing time for functional validation. I understand and respect your decision to publish this novel variant first.

Reply: Thank you.